**Subject Area:**
molecular biology/cellular biology

RNA-binding proteins, eukaryotic complexity, intrinsically disordered proteins, protein moonlighting, non-coding RNA

**Author for correspondence:**
Ewa Anna Grzybowska
e-mail: ewag@coi.waw.pl

†These authors contributed equally to the study.

# RNA–protein interactions: disorder, moonlighting and junk contribute to eukaryotic complexity

Anna Balcerak[1,†], Alicja Trebinska-Stryjewska[1,2,†], Ryszard Konopinski[1], Maciej Wakula[1] and Ewa Anna Grzybowska[1]

[1]The Maria Sklodowska-Curie Institute—Oncology Center, Roentgena 5, 02-781 Warsaw, Poland
[2]Biomedical Engineering Centre, Institute of Optoelectronics, Military University of Technology, Sylwestra Kaliskiego 2, 00-908 Warsaw, Poland

EAG, 0000-0003-3148-2439

RNA–protein interactions are crucial for most biological processes in all organisms. However, it appears that the complexity of RNA-based regulation increases with the complexity of the organism, creating additional regulatory circuits, the scope of which is only now being revealed. It is becoming apparent that previously unappreciated features, such as disordered structural regions in proteins or non-coding regions in DNA leading to higher plasticity and pliability in RNA–protein complexes, are in fact essential for complex, precise and fine-tuned regulation. This review addresses the issue of the role of RNA–protein interactions in generating eukaryotic complexity, focusing on the newly characterized disordered RNA-binding motifs, moonlighting of metabolic enzymes, RNA-binding proteins interactions with different RNA species and their participation in regulatory networks of higher order.

## 1. Introduction

With the possible exception of the unestablished first stages in the evolution of life, RNA has always been accompanied by some proteins. These proteins are necessary for RNA synthesis, maturation, transport, storage, regulation of stability and translatability. The RNA molecule from synthesis to degradation is constantly supervised and chaperoned by protein complexes, and their ever-changing protein content contributes to the regulation of the fate of the bound RNA. The scope of this regulation is crucial for the complexity of the organism. In bacterial cells, the regulation is relatively simple, with polycistronic operons controlling entire pathways. The discovery of CRISPR–Cas9 RNA-guided interference shows that bacteria also have at their disposal more sophisticated mechanisms involving RNA regulation employed in anti-phage defence [1]. However, real complexity is achieved in eukaryotes, where RNA regulation is a key factor, enabling levels upon levels of fine-tuning.

Recent efforts to identify new RNA-binding proteins (RBPs) in a screening technique called RNA interactome capture revealed that there might be about 1500–1900 RBPs in human cells [2], many more than previously estimated. Many of these proteins do not possess a canonical RNA-binding domain (RBD) and instead harbour an intrinsically disordered region (IDR), with RNA-binding potential. Structural disorder allows for more flexible and dynamic RNA binding, which contributes to the precision of the cellular response to stress and signalling. These new findings suggest that there is a whole new avenue of research to explore and that the role of RBPs in generating complexity may have been underestimated.

Another contribution to eukaryotic complexity is provided by dual- or multi-function proteins, which, under certain circumstances, may act as either

metabolic enzymes or regulatory RBPs, or possess additional biological roles. This additional layer of regulation enables RNA molecules to directly affect metabolic pathways.

The current perception of RNA–protein interactions is strongly biased towards a protein-centric approach, in which proteins regulate the expression and activity of RNA, not the other way around. Since only about 1–2% of the human genome encodes proteins and as much as 83–85% is transcribed [3], we have a large proportion of transcripts with unassigned functions, transcriptome 'dark matter'. The functions of these 'junk' transcripts are quite diverse, but mostly associated with fine-tuning of expression, providing a plethora of regulatory mechanisms, including direct regulation of protein activity. Thus, we have to acknowledge that RNA is not only a code that is subjected to the regulation, but also represents a potent regulatory factor itself.

Moreover, since the proportion of non-coding RNA rises with the complexity of the organism, it seems that its presence, along with the increased versatility of a functional proteome, is crucial for fine-tuned regulation. Thus, complex RBP–RNA regulatory networks controlling whole pathways in eukaryotic cells emerge as a new layer of complexity of which we still know very little.

In this review, we would like to discuss these new regulatory possibilities and present an emerging perspective of the importance and dynamics of RNA–protein interactions in the cell.

# 2. RNA-binding domains: ordered, disordered and unknown

## 2.1. Ordered, well-defined, globular RNA-binding domains

Until recently, it was assumed that RNA–protein interactions rely on well-defined RBDs of several types, each displaying a different mode of action, with different affinity and specificity. Most of these common RBDs interact with short sequences, but others recognize secondary structure features instead of specific sequence motifs. To ensure sufficient affinity, RBDs often work in concert, with multiple RNA-binding regions involved in specific binding. Notably, even these classic RBDs are very diverse, compared with DNA-binding domains: the three most abundant DNA-binding domains are present in 80% of all transcription factors, while the three most abundant RBDs are present in only 20% of all RBPs [4].

These well-structured RNA-binding motifs include RNA recognition motif (RRM), zinc finger, hnRNP K homology (KH) domain and double-stranded RNA-binding motif (dsRBM). All these motifs and their structures have been comprehensively described in many reports [5–9]. In this review, we will focus on other types of RNA binding, which may be a critical factor in eukaryotic complexity.

## 2.2. Intrinsically disordered RBDs

The relatively small number of genes in the human genome identified at the beginning of this century [10] came as a surprise to many researchers. It has become obvious that the complexity and diversity of humans is determined by factors other than the sheer number of genes. Alternative splicing

provided one answer, demonstrating that the same gene may encode various isoforms, possibly differing in function (for example, the long and short form of BCL-X have a different role in regulating apoptosis), but one of the other possible generators of functional complexity is the existence of RNA-binding intrinsically disordered proteins (IDPs), with pliable functionality. The properties and RNA-binding potential of these proteins are described below.

The use of a novel technique called interactome capture, which consists of cross-linking and immunoprecipitation (CLIP) coupled with subsequent mass spectrometry analysis, allowed for the identification of many new RBPs [11]. *In vivo* cross-linking of RNA–RBP interaction in CLIP enables identification of not only the RNA target molecule, but also the specific sequence to which the binding occurs. Different variations of CLIP (HITS-CLIP, iCLIP, eCLIP, PAR-CLIP, CRAC, uvCLAP) have been described in many reports and reviews [12–16].

A recent count of experimentally validated RBPs indicated there were about 1500, which represents 7–8% of all proteins [2,4]. Fifty-five per cent of these RBPs do not contain any known RBD [17]. Twenty-seven per cent do not contain canonical RBD and do not have any known function in RNA biology [18]. Among these new RBPs, a substantial group consists of IDPs, which lack a well-defined three-dimensional structure and can bind RNA using different motifs from classical RBDs. Table 1 displays selected examples of functionally important intrinsically disordered RBDs and their respective RNA-binding partners.

*Structural and physico-chemical features of RBPs*. A protein is considered intrinsically disordered when it possesses a region classified as disordered (an IDR) of more than 30 amino acids [19]. IDPs can display the whole spectrum of disorder, from relatively well organized, with known RBDs and disordered only in parts forming elastic linkers, to completely disordered proteins, which never adopt a single conformation. In the last case, IDPs form so-called fuzzy complexes analogous to the mathematical term 'fuzzy logic' [20]. Unstable conformation enables them to scan many structural possibilities and dynamically engage in the binding. This would allow a fast response to changing conditions in the cell.

Typical features of IDPs include low overall hydrophobicity and a large net charge, important for electrostatic interactions [21]. IDP conformation is dynamic and flexible, allowing for binding of multiple partners, but this binding is usually quite weak [22]. Low-affinity binding may be compensated for by multiplication of the binding sites. Even for well-structured RBDs, the affinity of the single domain is often insufficient for specific binding, thus the modular structure and multiple RNA-binding sites are fairly common in RBPs.

In spite of low affinity, IDPs may bind with high specificity, which is achieved by a large and highly complementary binding interface, whose formation is dependent on structural flexibility and the coupling of folding and binding [22].

*Induced folding in RBPs*. In the native state, disordered regions adopt an extended conformation, without structural features, but upon ligand binding (e.g. another protein, signal molecule, metal ion, nucleic acid) they may undergo induced folding, called disorder-to-order transition [23]. Database analyses revealed that structures of RNA-binding chains in RBPs are significantly less stable than DNA-binding or protein-binding protein chains, which suggests that most of the RNA-bound protein structures found in the Protein Data Bank (PDB) may be unstable in the unbound form,

royalsocietypublishing.org/journal/rsob   Open Biol. 9: 190096

**Table 1.** Selected examples of intrinsically disordered RBDs and their respective RNA partners.

| cellular structure | IDPs | type of RNA involved |
| --- | --- | --- |
| ribosome | L3, L4, L13, L20, L22, L24, L24e, S12, S14 | rRNA, tRNA, mRNA |
| | 4EBP1 | |
| spliceosome | ASF/SF2, SRp75, SRSF1 | mRNA, snRNA |
| P-granules | LAF-1 (RNA helicase) | mRNA |
| | MEG1, MEG3 | |
| decapping complex | DCP1, DCP2 | mRNA |
| P-bodies | EDC3, DHX9, XRN1 | mRNA |
| stress granules | FUS [9], EDC3 | intron RNA, lncRNA |
| RNA degradosome | RNase E | mRNA |
| nucleus | p53—signal integration hub | lncRNA |
| | SLBP | histone mRNA |
| | hnRNP A1, GroEL | hnRNA, mRNA |
| | PRC2 complex | lncRNA |
| | mediator complex | lncRNA |
| | NF-κB | lncRNA |
| | glucocorticoid receptor (GR) | lncRNA |
| | STAT3 | lncRNA |

and are stabilized by binding to RNA [23]. Examples of RNA–protein co-folding include human fragile X mental retardation protein (FMRP) [24], HIV-1 protein Rev [25] or DCL1 ribonuclease of *Arabidopsis thaliana* [26]. It may be hypothesized that, when a highly structured RNA is involved in the binding, this transition occurs with entropy transfer, resulting in partial unfolding of the RNA. Alternatively, post-translational modification, like methylation or phosphorylation, may impose structural change and modulate protein–RNA affinity. For example, phosphorylation of SRSF1 splicing factor triggers protein folding in the arginine/serine (RS)-rich region [27].

*Disorder in RBPs is evolutionarily conserved*. It has been demonstrated that RBPs are substantially enriched in IDRs and that the disorder is evolutionarily conserved [23,28]. Overall, as much as 20% of mammalian RBPs are disordered by 80% [28]. Moreover, the disorder is conserved, even when the underlying amino acid sequence is not (flexible disorder), which emphasizes the functional importance of the regions without a structure. The conservation is stronger in regions which have direct contact with RNA, and in this case conservation usually pertains to both sequence and structure (constrained disorder) [23]. Additionally, many of the well-characterized RBPs with known, classic RBDs can also be classified as intrinsically disordered, and while, in this case, the disordered region is not directly involved in the interaction with RNA, it still has an effect on the binding efficiency. For example, flexible linker regions separating RRMs in polypyrimidine tract binding protein 1 (PTBP1) influence RNA binding [29].

*Regulatory abilities of disordered RBPs*. IDPs display fascinating regulatory abilities. Even their taxonomic distribution (2% in Archaea, 4% in Eubacteria, 33% in Eukaryota and 44% in humans [30]) indicates that they are more abundant in complex organisms, suggesting their role in fine-tuned regulatory circuits [21]. Accordingly, these proteins often constitute signalling hubs, with the potential to regulate whole pathways, while metabolic enzymes (usually well structured) are underrepresented in this group [30].

Conformational sampling performed by IDRs enables a dynamic response to signals and expands regulatory possibilities of the system. It is worth noting that, even in the case of the known, well-defined RRMs, some conformational plasticity may be required for regulation. For example, in CUG-binding protein 2, which regulates the COX-2 transcript by binding to its 3′ untranslated region (UTR), the RRM was shown to exist in distinct substates, enabling a conformational switch between low-affinity binding, associated with dynamic RNA scanning, and high-affinity binding with RNA target locking [31]. This example additionally underscores the importance of conformational plasticity in fine-tuning of RNA discrimination.

### 2.2.1. Disordered RNA-binding motifs

Disordered RNA-binding motifs are recognizable patterns of disordered amino acid residues which could occur in a modular manner in RBPs and in some cases combine with globular domains. These motifs potentially cooperate with classical RBDs and play diverse biological roles, one of which is RNA binding [32]. They include short linear motifs (SLiMs), RG[G] repeats, RS/RG-rich, K/R patches, molecular recognition features (MoRFs) and LC sequences.

*Short linear motifs*. SLiMs are composed of up to 10 amino acid residue motifs located predominantly outside protein domains. They bind RNA with low affinity, but the specificity of the binding is accomplished by their multiplication [33,34]. SLiMs may also undergo post-translational modifications, which may change the specificity of the binding, so they can act as molecular switches [33].

*RG[G] repeats*. Motifs rich in arginine (R) and glycine (G) consist of at least three RG/RGG repeats, separated by 10 amino acid residues. This motif constitutes the second most

common RBD in the human genome. RNA-binding properties of this motif are poorly defined; it is known to have a broad, degenerate specificity in RNA recognition. However, the binding can be quite specific: for example, the RGG/RG domain of FMRP binds tightly to the RNA-containing G-quadruplex structure, acting as a specificity determinant [24]. RG[G]-containing proteins regulate RNA metabolism on all levels [35].

*RS/RG-rich.* RS-rich, RG-rich and other basic sequences can mediate both specific and non-specific interactions with RNA. Disordered, arginine and serine (RS) repeat-containing regions occur in a number of human proteins referred to as SR proteins and SR-like proteins. The most important group of SR proteins are pre-mRNA splicing factors which bind to exonic splicing enhancers and stimulate the excision of adjacent introns. These proteins contain one or two RBDs at the N-terminus and a C-terminal arginine–serine-rich domain which also could directly bind RNA. Splicing efficiency depends on the length of the RS repeat and could be modified by phosphorylation of RS, which promotes a transition from intrinsically disordered to an arch-like structure, influencing RNA-binding properties. This phenomenon can be observed in serine/arginine-rich splicing factor 1 (SRSF1) and the RNA-helicase DDX23 [27,36]. Except for splicing, SR-rich proteins are also involved in other processes, namely export, translation and maintenance of genome stability.

*K/R basic patches.* This motif is composed of four to eight lysines (K) or arginines (R), which form a highly positive and exposed interface. K/R patches are highly abundant among non-canonical RBPs and can group at multiple positions within RBPs, frequently flanking globular domains and probably cooperating with them in RNA binding [28,32].

*Molecular recognition features.* Most MoRFs are up to 25 amino acid residues long, but some of them are 50 or more residues long [37]. An important feature of this type of motif is its ability to undergo dynamic disorder-to-order transition upon ligand binding owing to preexisting structure, predominantly α-helix [38]. MoRFs are classified as a-MoRFs, b-MoRFs and i-MoRFs (which form α-helices, β-strands and irregular structures, respectively, upon ligand binding) or could be mixed [37,39]. This motif provides unique specificity and diversity as well as reversibility of binding.

*Low-complexity sequences.* Low-complexity (LC) sequences contain up to 100 amino acids, and are composed of many repeats of the same amino acid or several amino acids [39]. Significant enrichment in these sequences occurs in regulatory proteins binding RNA and DNA [40]. LC sequences have been shown to be in a disordered state when the protein is soluble [41]. With increasing LC concentration, proteins containing these sequences (for example, FUS and hnRNPA2) can polymerize into amyloid-like fibres and undergo a reversible phase transformation to a hydrogel-like state. As elaborated below, LC motifs are characteristic of proteins that are part of RNA storage granules.

### 2.2.2. Disorder is important for the formation of RNA–protein functional units

The majority of RBPs (as well as RNAs) are localized in the nucleus. Export of the RNA to the cytoplasm is tightly controlled, requires many steps and concerns mostly mRNAs, rRNAs, tRNAs and miRNAs. There are also examples of long non-coding RNA (lncRNA) present in the cytoplasm, which is consistent with many of them being capped, poly-adenylated and undergoing splicing, thus fulfilling the criteria for efficient export. Nevertheless, compared with mRNAs, lncRNAs are significantly enriched in the nucleus [42]. Nuclear or cytoplasmic RNAs may be organized into big, stable, well-defined macromolecular complexes, with a specific set of associated proteins and particular functions like the spliceosome (splicing) or the ribosome (translation), or they can be present in many types of RNA granules, for which the functions assigned so far encompass RNA storage, transport (coupled with localized translation) or, possibly, degradation. In the formation of both types of these functional units, IDPs play critical roles. Figure 1 describes the localization and functional cooperation of these functional units in the eukaryotic cell.

*Spliceosome (nucleus).* Approximately half of the combined sequence of abundant spliceosomal proteins is predicted to be disordered. About 80% of spliceosomal proteins were predicted to contain at least one IDR, in contrast with the calculated fraction of about 35% for the entire human proteome [43]. It has been shown that disordered arginine and serine repeats in SR proteins are vital for spliceosome formation and functioning. RS repeats directly bind RNA (non-specific, sequence-independent binding), contribute to RRM binding affinity (specific RNA binding) or mediate protein–protein interactions (PPIs) [28]. Additionally, it was demonstrated that early spliceosomal proteins engaged in molecular recognition and dynamics are more disordered and evolutionarily younger than late proteins, which are directly involved in splicing catalysis. It has also been calculated that spliceosomal proteomes contain more intrinsic disorder than ribosomal proteomes [43].

*Ribosome (cytoplasm).* Intrinsic disorder is not only abundant in ribosomal proteins, but it is also necessary for their function and, as such, is highly conserved. According to the criteria set out by Gunasekaran et al. [44], almost all eukaryotic ribosomal proteins can be classified as disordered [45]. Ribosomal IDPs have specific features; about half of them are built like a tadpole, with a globular domain and an unstructured tail penetrating deeply into the ribosomal core and interacting with RNA. Upon interaction, disordered regions undergo disorder-to-order transition, co-folding with RNA. Many ribosomal proteins are also completely disordered in the unbound state and do not possess any globular domain [45]. Another particular feature of many ribosomal proteins is that they display off-ribosome functions [46].

*RNA granules.* The presence of disordered RBPs has been linked to the formation of membraneless cellular ultrastructures, rich with condensed proteins and RNA. The formation of these granules requires liquid–liquid phase separation (LLPS [47]). LLPS provides a quick method for sufficient condensation and separation of specific components in one place without the necessity for crossing a membrane. Additional features of RNA granules include relatively easy access by external factors and the possibility of fast dissolution, which makes the droplets highly dynamic. They have all the properties of a liquid, like droplet fusion, wetting and dripping [48]. These structures are fairly common and can be found in the nucleus and in the cytoplasm. Nuclear bodies include very well-known structures such as the nucleolus, Cajal bodies, and also paraspeckles

royalsocietypublishing.org/journal/rsob   Open Biol. 9: 190096

**Figure 1.** RNA–protein functional units in the cell rely on disordered RBPs. Most of the RNA is in the nucleus, where it is transcribed, spliced, exported and/or degraded, stored or engaged in chromatin regulation (lncRNA). Mature RNA, which is exported to the cytoplasm, can either be translated on the ribosome or also degraded, stored or transported to the site of localized translation (mostly in neurons or developing embryo). Different forms of cellular stress stall translation and freeze RNA–protein translation complexes in stress granules, until optimal conditions are restored or the cell dies. Cellular RNA can be bound in big macromolecular complexes (spliceosome or ribosome) or be stored, sequestered, transported or degraded in different types of RNA granules. In both cases, proteins forming these ribonucleic entities are significantly disordered and the disorder is crucial for their existence. EJC, exon-junction complex.

or promyelocytic leukaemia bodies. Cytoplasmic bodies may differ depending on cell type and include processing bodies (P-bodies), stress granules, RNA granules (neurons), P-granules (germ cells) and centrosomes. Different granules may vary in the extent of condensation; for example, in yeast cells, P-bodies are liquid-like, while stress granules are more solid-like. Interestingly, in mammalian cells, both P-bodies and stress granules are liquid-like [49].

It has been shown that LC sequences present in IDPs play a role in the formation of these structures and that high concentrations of LC sequences lead to a reversible phase transition to a solid hydrogel state [50]. LC sequences were found in many disordered proteins which participate in droplet formation (e.g. FUS, TDP-43, hnRNPA1 and TIA-1), but they are not absolutely necessary, since protein Tau, while disordered and highly charged, lacks true LC sequences, but is still capable of forming droplets [51].

The functions of RNA granules are not entirely clear, but they are likely to include RNA transport, storage, preservation and separation and a role in mRNA decay. They are also especially important in aggregation-associated diseases.

*Medical implications of LLPS.* A growing body of evidence indicates that the components of RNA granules with a potential to enhance liquid-to-solid phase transition can be responsible for aberrant RBP aggregation in neurons, observed in amyotrophic lateral sclerosis (ALS) and frontotemporal lobar degeneration (FTLD). Several mutations in genes encoding such IDPs as FUS, hnRNPA1, TARDBP (TDP-43) and TIA-1 have been linked to these age-related neurodegenerative diseases. Many of the identified mutations result in a replacement of prion-inhibiting charged amino acids with neutral or prion-promoting ones (aromatic, hydrophobic). Indeed, *in vitro* and *in vivo* studies showed that mutated RBPs show a higher propensity for transition from the liquid and reversible hydrogel state to irreversible fibrillar assemblies. This process is usually mediated by a mutated LC domain, but changes in the other parts of the protein can also indirectly increase its ability to undergo phase transition [52,53]. The pathogenic significance of RBP aggregates in ALS and FTLD is still not entirely clear. The aggregates themselves can be toxic to the cells, as in the case of the amyloids. Another possibility, not necessarily mutually exclusive, is that the poorly soluble assemblies can selectively trap RBPs,

impairing their subcellular localization and function. This would disrupt RNA granule turnover and transport in neurons, affecting localized protein synthesis and causing neurotoxicity [54].

## 3. Metabolic enzyme moonlighting: binary structural switch

Metabolic enzymes constitute a class of proteins with well-defined structural features, crucial for enzymatic activity and—accordingly—proteins with the least percentage of disordered regions. There has been a general opinion that they perform only one, highly specific function, but in some cases, described below, they have another function, which is often related to RNA binding. Therefore, these enzymes may 'moonlight', performing some tasks physiologically distinct from their role in basic metabolism. Thus, moonlighting of metabolic enzymes provides yet another explanation for the small number of genes in the human genome and adds to the complexity of eukaryotic regulation. Owing to their promiscuous binding, IDPs were initially hypothesized to be good candidates for moonlighting [55], but, to date, analysis of interactomes does not support this notion [56]. This is consistent with the fact that the moonlighting pertains mostly to metabolic enzymes, which are generally very well structured. Thus, in contrast with intrinsically disordered RBPs, they are able to switch only between two conformations.

A well-known example of dual-function protein moonlighting as RBP is aconitase, an enzyme that catalyses the interconversion of citrate to isocitrate. Aconitase is an iron–sulfur protein and, in normal conditions (unchanged iron level), it functions as an enzyme. Its moonlighting as an iron regulatory protein 1 (IRP1) has been described many times [57–59], and it includes binding to the UTRs of specific transcripts (encoding ferritin and transferrin) to regulate iron intake in response to iron deficiency. When the iron level is low, the iron–sulfur cluster disintegrates, which destroys the active site of the enzyme and shifts the protein to an open conformation, which enables RNA binding. Only the cytoplasmic form of actonitase (encoded by ACO1) moonlights as an iron regulatory protein.

Another well-known example of a metabolic enzyme acting as an RBP is glyceraldehyde-3-phosphate dehydrogenase (GAPDH), a glycolytic enzyme with a plethora of extra-glycolytic functions, which include its RNA-binding properties [60,61]. GAPDH binds RNAs of many kinds, including tRNA, numerous mRNAs, rRNA and TNF-α hammerhead ribozyme [61,62]. While there is no doubt that it has RNA-binding potential, and despite the fact that it is well structured and its structure is known, hitherto no RBD has been characterized in GAPDH and, thus, it falls into a group of non-canonical RBPs. It has been suggested that RNA binding requires a large portion of the protein, involving the Rossman fold (nucleotide-binding structural motif), positive groove and dimer interface, but the precise mechanism remains elusive [62].

Aside from aconitase and GAPDH, RNA binding by metabolic enzymes has been reported many times (reviewed by Ciesla [62]) and includes enzymes involved in glycolysis and the pentose cycle (aldolase, lactate dehydrogenase, phosphoglycerate kinase, glucose-6-phosphate dehydrogenase), the tricarboxylic acid cycle (glutamate dehydrogenase and isocitrate dehydrogenase) or thymidylate synthesis (thymidylate synthase and dihydrofolate reductase). Additionally, recent RNA interactome capture approaches have provided 23 new examples of metabolic enzymes identified as RBPs [63].

Identification of such a relatively large group of bi- or multi-functional enzymes, with RNA-binding properties, has led to the formulation of the RNA–enzyme–metabolite (REM) hypothesis, according to which moonlighting metabolic enzymes mediate the connections and, possibly, cross-talk between cellular metabolism and gene expression [64].

## 4. Next tier of regulation: RBPs and their RNA targets compose eukaryotic regulatory networks

Regulation at the RNA level was traditionally understood in terms of RBP-mediated control over transcript quality/stability, so only coding sequences were considered and only in the context of regulation by proteins. However, non-coding RNA constitutes the prevailing part of eukaryotic transcriptomes, especially in more complex species. It is becoming more and more obvious that this huge transcriptional effort has a function and these RNAs are not junk or molecular fossils, but functional elements of sophisticated regulatory circuits.

There are different types of non-coding RNAs, including well-known RNAs with specific functions, such as rRNAs or tRNAs in translation, snRNAs involved in splicing, nucleolar snoRNAs, which guide chemical modifications of other RNAs, and, finally, the two big classes of regulatory ncRNAs: small non-coding RNAs (microRNAs of approx. 21–24 nt, responsible mostly for transcript silencing, and piRNAs of approx. 26–31 nt, involved in transposon silencing) and lncRNA (more than 200 nt) with a growing list of different functions, including the regulation of chromatin accessibility and transcription, miRNA sponging, pathway regulation or direct regulation of specific proteins. The non-coding part also includes introns, which constitute most of the eukaryotic pre-mRNA, and their functionality is still enigmatic.

As demonstrated in figure 2, the proportion of non-coding RNAs rises with the complexity of the organism. Interestingly, it can be observed that, while the number of genes for lncRNAs is dramatically higher in mouse and human, the number of RBPs in eukaryotes is relatively stable across the species. While the numbers in figure 2 might be a little biased depending on the number of datasets included (RBP count in species corresponding to a low number of datasets may be underestimated), the results for yeast, mouse and human are very sound (respectively: 8, 6 and 6 datasets) and the number of RBPs is similar. There is a huge leap between bacterial RBPs (estimated at about 180 proteins) and a 10-fold increase in eukaryotes, but, among eukaryotes, the numbers are relatively stable. This suggests that a similar number of RBPs is responsible for binding a significantly higher number of transcripts. How is this achieved? One explanation may include protein plasticity brought about by intrinsic disorder, which enables one RBP to bind many RNA targets. Indeed, it was observed that the

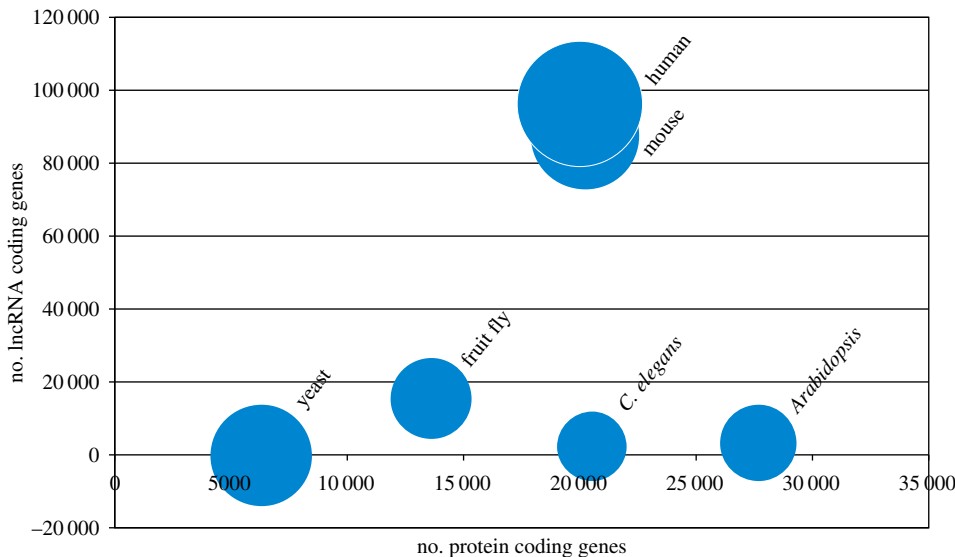

**Figure 2.** The number of lncRNA is much higher in more complex eukaryotes, while the number of RBPs remains similar. Bubble plot showing the proportion of protein coding to lncRNA coding genes across species. The number of lncRNA coding genes from the NONCODEV5 database [65]. The bubble size denotes the number of RBPs [2]. Number of datasets: yeast (*Saccharomyces cerevisiae*)—8, human (*Homo sapiens*)—6, mouse (*Mus musculus*)—6, *Arabidopsis thaliana*—3, fruit fly (*Drosophila melanogaster*)—2, *Caenorhabditis elegans*—1.

number of RNA targets for a single eukaryotic RBP can be as high as several tens of thousands [66].

Thus, in this part of the review, we would like to address the changing perspective on RBP targets, their genomic distribution, newly assigned functions and their role in building eukaryotic regulatory networks.

## 4.1. Genomic distribution and newly discovered functionalities of RBP binding targets: growing importance of non-coding RNAs

Analysis of the genomic distribution of protein-binding RNA motifs revealed that a large proportion of these motifs is localized within non-coding parts of the genome. The most popular regions include introns and the 3′UTRs of mRNA [67,68]. This distribution of motif occupancy is quite understandable since non-coding regions of the transcripts are less constrained in protein binding than the coding regions and 5′UTRs, in which abundant protein binding might interfere with function.

### 4.1.1. RBP binding to intronic sequences

Intronless genes constitute only a small and, surprisingly, quite evolutionarily recent part of eukaryotic genomes [69]. Introns obviously contribute to eukaryotic complexity in many ways, not only enabling the variability of alternative splicing but also directly influencing transcription (intron-mediated enhancement [70]), contributing to the regulation of transcript stability and nuclear export. Splicing is in fact crucial for efficient export of the mature mRNA to the cytoplasm, since these processes are functionally coupled with splice factors facilitating access to the export machinery [71]. Moreover, introns and splicing are vital for nonsense-mediated decay, owing to exon-junction complex positioning on mRNA. Introns are also known to host many coding sequences, especially encoding for miRNA [72] or snoRNA [73]. The abundance of RBP binding to intronic sequences

suggests that these known properties still do not cover the whole spectrum of introns' functional importance.

Depending on the function of RBPs, the proportion of intron binding may vary from around 10–20% for proteins involved in mRNA transport and translation or mRNA degradation (those proteins bind mostly to the sites localized in the 3′UTR) to more than 80% in splicing, polyadenylation and RNA processing-associated proteins [67]. Examples include the protein Aquarius (AQR, also called IBP160), which couples pre-mRNA splicing and snoRNP (small nucleolar ribonucleoprotein) biogenesis [74] and FUS protein, which regulates splicing of genes coding for RBPs by binding to their highly conserved introns [75]. Notably, besides FUS, some other RBPs involved in the pathology of ALS also bind to intronic sequences, like splicing factor proline and gluta-mine-rich (SFPQ) or TDP-43. SFPQ transcript displays intron retention, and SFPQ protein binds to the retained intron of the SFPQ transcript, autoregulating the expression [76]. TDP-43, a potent regulator of mRNA processing and another very well-known factor in ALS pathogenesis, was also shown to bind within the exceptionally long introns of its pre-mRNA targets [77].

Protein HuR, which is usually perceived to be involved in post-transcriptional regulation, was also shown to bind abundantly to intronic sequences [68], suggesting that it is in fact engaged in pre-mRNA processing, and not only in the control of mature mRNA stability.

While most intronic sequences are considered to degrade shortly after incision, some persist either in a form of sequences retained in mRNA or as a stable intronic sequence RNA (sisRNA), a new class of non-coding RNA identified in *Xenopus* oocytes [78]. Functional intronic sequences tend to be more stable and more conserved. For example, colorectal neoplasia differentially expressed lncRNA (CRNDE) promotes glycolysis in cancer cells by modulating the level of GLUT4. It has been shown that the CRNDE-encoding gene contains a highly conserved intron sequence (gVC-In4), involved in the regulation of glucose metabolism in cancer cells [79].

### 4.1.2. 3'UTR-mediated regulation: miRNA competition with RBPs and co-translational protein assembly

In addition to conventional control executed by RBPs on RNA metabolism (transcription, maturation, export, storage, stability, quality control, degradation, spatial and temporal regulation of translation), there is some new evidence showing that they have an impact on microRNA-regulated decay. While their participation in miRNA processing is obvious, including maturation controlled by microprocessor, export, cleavage by Dicer and RISC-assisted silencing, there are some reports pointing to their role in the regulation itself, not only as molecular machines that execute the programme but also as a part of the decisive switch.

It has been known for a long time that mRNA stability can be controlled by RBP binding to the 3'UTR of the transcripts [80–83]. Some specific 3'UTR motifs have been characterized as regulatory, including AU-rich regions or specific structural motifs. The AU-rich sequence (ARE), UAUUUAU, is present in many transcripts encoding signalling factors, including cytokines, growth factors and oncogenes [84]. Among the proteins that can bind to AREs are HuR/ELAV [81], tristetraprolin (TTP) [85], AUF [86] and FXR1 [87]. Some of these ARE-binding proteins (ARE-BPs) promote degradation of the target transcript, while others, like the HuR family of proteins, mostly cause stabilization of the targeted message.

Since 3'UTRs turned out to be the primary target of microRNA-mediated regulation, it is only logical to assume that there might be some competition (or cooperation) between classical RBPs and miRNAs. While only a comprehensive and exhaustive analysis of coincidence in binding of RBPs and miRNAs to specific 3'UTR targets can decisively prove the scope of these interactions, there is some evidence to support its existence. It has been shown, using RIP-chip and PAR-CLIP, that the well-characterized RBP HuR binds to many of the same targets (75%) as protein Ago (RISC complex component) [88], suggesting a competition for the binding site. Indeed, many specific examples of HuR competition with miRNAs have been described, including the regulation of the cationic acid transporter 1 (CAT1) mRNA (miR-122), topoisomerase IIα (TOP2A) mRNA (miR-548c-3p), nucleolin mRNA (miR-494) or erb-B2 receptor tyrosine kinase 2 (ERBB2) mRNA (miR-331-3p) [89]. In all of these examples, HuR binding stabilizes the messenger, preventing miRNA-mediated silencing. There are also examples of cooperation, such as in the case of tristetraprolin (TTP), which destabilizes ARE-containing mRNAs (mRNAs rich with AU sequences) by promoting miR-16-mediated silencing due to physical interaction with RISC [90]. Interestingly, HuR can also act as a destabilizing factor, promoting silencing by recruiting the transcript to RISC (let-7, c-Myc transcript) [91].

Recently, another potential role of the 3'UTR in eukaryotic regulation has emerged. In prokaryotes, the assembly of the most protein complexes occurs co-translationally by sequential translation of the polycistronic mRNA. It has been shown that, in eukaryotic organisms, the assembly of many protein complexes also occurs co-translationally [92]. The authors used selective ribosome profiling to show that ribosomes which synthesize new protein often are in a complex with another protein. This other protein assists co-translationally in proper folding of the nascent polypeptide, and, interestingly, represents another subunit of the same protein complex as the first protein. Disruption of this process results in protein aggregation. How does this co-translational assembly occur? Mayr [93] suggests that the proximity of the two subunits near translating ribosomes can be achieved either by a protein which bridges two mRNAs, so the translation of both subunits can occur in the same localization (alternatively, the mRNAs can be held together in RNA granules, without direct bridging) or—more probably—the bridging is mediated by the 3'UTR of the mRNA which is actively translated. This 3'UTR recruits fully folded subunits via a multi-functional RBP, enabling co-translational folding. Thus, we are a step closer to characterizing the 'eukaryotic operone' the existence of which has been postulated for a long time.

### 4.1.3. RBP–lncRNA interactions regulate transcription

LncRNAs are probably the most promising source of eukaryotic complexity, since, to date, only a small fraction of them have an assigned function and their regulatory abilities seem to be vastly underappreciated. The ENCODE project has annotated about 16 000 lncRNA genes (28 000 distinct transcripts) in humans [94]. NONCODEV5 database estimations are even higher [65]. LncRNAs are involved in the regulation of expression on various levels, including chromatin remodelling, regulation of transcription, epigenetic mechanisms and the activity of transcriptional enhancers, but also on the post-transcriptional level, in the regulation of splicing, RNA stability and translation. Most of these activities take place in the nucleus, but, since lncRNAs share common mRNAs features like a 5' 7-methylguanosine cap and a 3' poly(A) tail, they can also be targeted to the cytoplasm. Moreover, lncRNAs are expressed in a tissue- and context-specific manner [95], which creates many opportunities for the specific regulation and fine-tuning of cellular processes.

To date, functions have been assigned to many lncRNAs, but specific RNA–protein interaction has not been described in all cases. Some of these confirmed interactions are listed in table 2.

One of the most interesting examples of lncRNA regulation is polycomb repressive complex 2 (PRC2), a histone methyltransferase active in epigenetic silencing during development and carcinogenesis [96,101]. It binds with low affinity hundreds of cellular RNAs, but it does not possess any known RNA-binding motif. Among many others, PRC2 was shown to bind important functional lncRNAs, including HOTAIR, Xist, RepA, Kcnq1ot1, Braveheart, MALAT1, H19, ANRIL and MEG3 [96]. LncRNA has been demonstrated to scaffold PRC2, which helps to assemble histone modification enzymes [102]. It has been hypothesized that massive lncRNA binding to polycomb proteins represents a mechanism by which PRC2 is recruited to chromatin, and by which epigenetic silencing is regulated. Several models have been proposed to explain this activity [96,103], and, while they are still disputed, it seems clear that RNA binding by PRC2 represents yet another example of the appearance of low-order (low-affinity binding) mask hidden complexity.

### 4.2. Eukaryotic regulatory networks

The existence of regulatory units of higher order in eukaryotic cells was proposed many years ago [104]. Recent results of

royalsocietypublishing.org/journal/rsob    Open Biol. 9: 190096

**Table 2.** Selected examples of RBPs and their target lncRNAs.

| protein | lncRNA | cellular process | reference |
|---|---|---|---|
| DNMT1 | DACOR1 | DNA methylation and transcriptional | [75] |
| DNMT3b | MEG3 | repression | [76] |
| PRC2 complex | HOTAIR, Xist, RepA, Kcnq1ot1, Braveheart, MALAT1, H19, ANRIL, MEG3 | epigenetic silencing during development and carcinogenesis | [96] |
| NFκB | NKILA | regulation of phosphorylation status of | [77] |
| STAT3 | lnc-DC | transcription factors | [78] |
| p53 | DINO | regulation of the DNA-damage-induced p53 response | [79] |
| mediator complex | ncRNA-a3, ncRNA-a7 | transcriptional co-activation | [97] |
| | | histone H3 phosphorylation | [98] |
| GR | GAS5 | suppressing the expression of the glucocorticoid-responsive genes | [99] |
| AR, ER, PR, GR, thyroid hormone receptor, RAR | SRA RNA | co-activation of hormone receptors | [100] |

high-throughput methods combined with computational efforts to characterize RBPs and their binding targets allow the formulation of more specific hypotheses of how these regulatory networks operate.

First predictions of RNA–protein interactions were based on amino acid sequence similarities and the presence of the known RBDs (e.g. RNApred [70]). In 2011, the Tartaglia group introduced the catRAPID server, able to predict RNA–protein interactions relying on physico-chemical properties (secondary structure, hydrogen bonding, van der Waals forces) instead of sequence patterns [71]. With the growing number of newly discovered RBPs with unconventional RBDs, this computational approach has proved to have a larger prediction scope. Using catRAPID and *ex vivo* data, the Tartaglia group was able to probe the expression patterns of about 1000 human RBPs and their mRNA targets, revealing that, for the predicted mRNA–RBP pairs, the expression patterns were strongly correlated or anti-correlated [72]. Interestingly, positive correlation patterns were related to genes encoding for proteins involved in proliferation and cell cycle control, while negative correlation patterns were related to survival, growth and differentiation. Nishtala *et al.* [66] performed an analysis on CLIP data for 60 human RBPs and also reported similar regularities in expression patterns. Additionally, they observed that the co- or anti-expression is stronger on the protein level (approx. 95% association) than on the transcript level (approx. 78%), pointing to the role of post-transcriptional regulation [66]. RNA–RBP regulatory networks seem to operate as complex units in which multiple RBPs interact with each other, providing either cooperative target regulation (e.g. DDX3X and CARIN1 in control of RAC1 translation) or competitive target regulation (e.g. PABPC1 and YBX1 antagonistically controlling YBX1 mRNA) [73].

It was observed that proteins which interact with multiple RBPs are frequently RBPs themselves [67]. Thus, the proportion of RBPs in PPI networks predict whether a given protein is a candidate RBP. The Yeo group has developed an algorithm termed Support vector machine Obtained

from Neighborhood-Associated RBPs (SONAR), which is able to identify an RBP using information from PPI networks, without relying on its sequence or structure [67]. This has led to the identification of hundreds of previously unannotated RBPs across multiple species (1923 proteins predicted in humans).

These new computational approaches should allow further characterization of the RNA–RBP regulatory networks, which control whole processes and pathways in an intricate and largely unknown manner.

## 5. Conclusion

Considering the emerging scope of the regulatory effort in complex Eukaryota, it appears that what we currently perceive is only the tip of the iceberg. With the advent of reliable high-throughput techniques, enabling analysis of whole transcriptomes and proteomes, it transpires that the number of RBPs is much higher than previously estimated and that their RNA-binding modes may differ substantially—from RBPs with clearly defined, structured RBDs, moonlighting proteins with dual specificity, to unstructured, multitasking IDPs. Of particular interest is the situation in which the protein does not possess any known RNA-binding motifs, but still binds a large number of RNAs with low affinity. As can be seen in the example of polycomb, such a binding mode may have profound regulatory consequences. Specific binding with low affinity is characteristic of RNA-binding IDPs, and their role in RNA regulation is increasingly being recognized. In the past few decades, we have observed an important paradigm shift—from the static image, in which only well-folded proteins with defined domains were able to interact and function—to a dynamic, far more complex picture, where conformational plasticity is required for full functionality. Intrinsic disorder is crucial for this plasticity and, as it appears, for the formation of RNA–protein functional units, in the form of either macromolecular complexes or RNA granules.

Another change of perspective considers the role of RNA in ribonucleocomplexes. From Thomas Cech's discovery of

royalsocietypublishing.org/journal/rsob   Open Biol. **9**: 190096

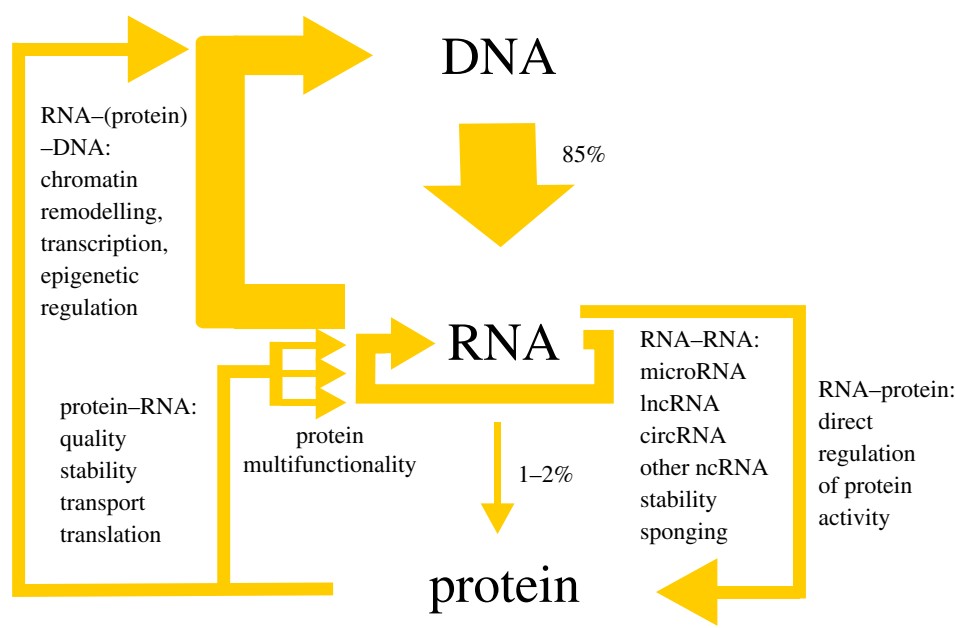

**Figure 3.** RNA occupies a central position in cellular regulatory circuits. Transcription of RNA, its quality, stability and translation are regulated by proteins. However, the central dogma in which DNA encodes proteins via RNA needs to be updated, taking into account the role of RNA as a main player in the regulation of expression, not only as a messenger. Almost all DNA is transcribed, but only 1–2% encodes proteins. At the same time, the estimates of the ENCODE consortium [105] assign biochemical function to 80% of the genome. Thus, the activity of the non-coding part of the genome encompasses the regulation of the RNA itself (microRNA, lncRNA and the other types of ncRNA), regulation of DNA (chromatin architecture, transcription, epigenetic modifications) and direct regulation of protein activity. As stressed in the article, eukaryotic RBPs have multiple functions.

catalytic RNAs [74], it is known that this molecule can be quite active, but it was still mostly perceived as a code or a passive scaffold. This outlook is also about to change, especially with growing knowledge about the role of non-coding RNA. The large majority of RBPs bind to non-coding RNA, or, in the case of mRNAs, to their non-coding regions (UTRs). A large proportion of transcribed, non-coding RNA in eukaryotic genomes, especially in the context of its taxonomic distribution, with the highest proportion in the most complex organisms, indicates its central role in the regulation of complexity. As depicted in figure 3, RNA plays a decisive role in the fine-tuning of expression by performing different tasks that include: (i) control of transcription and epigenetic factors (RNA–DNA regulation), (ii) self-control (RNA–RNA regulation; splicing, stability, translatability), and (iii) direct regulation of RBPs' activity

(RNA–protein regulation). This regulatory effort is orchestrated by RNA–RBP complexes, which work in concert, forming complicated networks.

Overall, as stressed in this review, formerly unappreciated features such as structural disorder or non-coding properties play a crucial role in shaping eukaryotic regulatory networks and, as a result, eukaryotic complexity.

Data accessibility. This article has no additional data.

Authors' contributions. A.B., A.T.-S. and M.W. edited selected paragraphs, R.K. participated in the preparation of the table and figures, E.A.G. designed and edited the whole text and participated in the preparation of figures.

Competing interests. We have no competing interests.

Funding. This work was supported by the Polish National Science Center (grant nos. 2014/14/M/NZ1/00437 and 2015/17/N/NZ5/01392).

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
