## [Reviewer comments · Open Biology]

Review History

RSOB-18-0236.R0 (Original submission)

Review form: Reviewer 1

Recommendation

Major revision is needed (please make suggestions in comments)

Are each of the following suitable for general readers?

- a) **Title**
No
- b) **Summary**
Yes
- c) **Introduction**
Yes

Is the length of the paper justified?

No

Should the paper be seen by a specialist statistical reviewer?

No

Is it clear how to make all supporting data available?

Not Applicable

Is the supplementary material necessary; and if so is it adequate and clear?

Not Applicable

Do you have any ethical concerns with this paper?

No

Comments to the Author

The authors chose a heroic topic "biology complexity" as the center discussion for this mini-review: intrinsic disordered region, moonlighting enzyme and non-coding RNA. The ambition of the authors are respectable. However, each topic has been the center of investigation of hundreds of laboratories in the world. For each topics, there has been decades of studies with the recent emerge of new findings that has shifted the paradigm. Thus, it is impossible to conduct an in-depth review on these hot topics in a short mini-review. This article glossed on the surface of these three important and sophisticated topics but didn't have a chance to provide the insight the reader would appreciate. The references in the review is mostly outdated and the figures provided are not particular original. Instead of a pan-spectrum of many mechanisms that accounts for biological complexity (which is easily a title for a multi-volume book), maybe a better strategy is to focus on one of the topics, read through the recent key findings and write a review on that.

Review form: Reviewer 2

Recommendation

Accept with minor revision (please list in comments)

Are each of the following suitable for general readers?

- a) **Title**
Yes
- b) **Summary**
Yes
- c) **Introduction**
Yes

Is the length of the paper justified?

Yes

Should the paper be seen by a specialist statistical reviewer?

No

Is it clear how to make all supporting data available?

Not Applicable

Is the supplementary material necessary; and if so is it adequate and clear?

Not Applicable

Do you have any ethical concerns with this paper?

No

Comments to the Author

This is a very good, very thorough review that merits publication in Open Biology.

I have made comments and a large number of corrections on the original MS word manuscript. Most of these corrections are minor - mainly grammatical or stylistic. The authors should review these and make sure that my changes are consistent with what they have intended to communicate.

Decision letter (RSOB-18-0236.R0)

28-Jan-2019

Dear Dr Grzybowska,

We are writing to inform you that your manuscript RSOB-18-0236 entitled "RNA-protein interactions: disorder, moonlighting and junk contribute to eukaryotic complexity" has, in its current form, been rejected for publication in Open Biology.

The referees have recommended that major revisions are necessary but that the manuscript has potential; hence, we would like to actively encourage you to revise the manuscript accordingly, and resubmit. Nevertheless, please note that this is not a provisional acceptance.

The resubmission will be treated as a new manuscript and will re-enter the review process. Every attempt will be made to use the original referees, but this cannot be guaranteed. Please note that resubmissions must be submitted within six months of the date of this email. In exceptional circumstances, extensions may be possible if agreed with the Editorial Office. Manuscripts submitted after this date will be automatically rejected.

Please find below the comments made by the referees, not including confidential reports to the Editor, which I hope you will find useful. If you do choose to resubmit your manuscript, please upload a 'response to referees' document including details of how you have responded to the comments, and the adjustments you have made.

To upload a resubmitted manuscript, log into <http://mc.manuscriptcentral.com/rsob> and enter your Author Centre, where you will find your manuscript title listed under "Manuscripts with Decisions." Under "Actions," click on "Create a Resubmission." Please be sure to indicate in your cover letter that it is a resubmission, and supply the previous reference number.

Sincerely,

The Open Biology Team

mailto: openbiology@royalsociety.org

Editor's Comments to Author(s):

Although one referee is supportive, the comments of the second suggest a need to expand the review to cover the areas in more depth. If you would wish to do this, please get back to me with a plan and we can move forward

David Glover

Associate Editor,

Reviewer(s)' Comments to Author(s):

Referee: 1

Comments to the Author(s)

The authors chose a heroic topic "biology complexity" as the center discussion for this mini-review: intrinsic disordered region, moonlighting enzyme and non-coding RNA. The ambition of the authors are respectable. However, each topic has been the center of investigation of hundreds of laboratories in the world. For each topics, there has been decades of studies with the recent emerge of new findings that has shifted the paradigm. Thus, it is impossible to conduct an in-depth review on these hot topics in a short mini-review. This article glossed on the surface of these three important and sophisticated topics but didn't have a chance to provide the insight the reader would appreciate. The references in the review is mostly outdated and the figures provided are not particular original. Instead of a pan-spectrum of many mechanisms that accounts for biological complexity (which is easily a title for a multi-volume book), maybe a better strategy is to focus on one of the topics, read through the recent key findings and write a review on that.

Referee: 2

Comments to the Author(s)

This is a very good, very thorough review that merits publication in Open Biology.

I have made comments and a large number of corrections on the original MS word manuscript. Most of these corrections are minor - mainly grammatical or stylistic. The authors should review these and make sure that my changes are consistent with what they have intended to communicate.

Author's Response to Decision Letter for (RSOB-18-0236.R0)

See Appendix A.

RSOB-19-0096.R0

Review form: Reviewer 1

Recommendation

Accept with minor revision (please list in comments)

Are each of the following suitable for general readers?

- a) **Title**
Yes
- b) **Summary**
Yes
- c) **Introduction**
Yes

Is the length of the paper justified?

Yes

Should the paper be seen by a specialist statistical reviewer?

No

Is it clear how to make all supporting data available?

No

Is the supplementary material necessary; and if so is it adequate and clear?

No

Do you have any ethical concerns with this paper?

No

Comments to the Author

Please see the attached file (below), since it will be easier for the authors to read.

The authors, as they indicated, made most of the corrections that I had previously suggested. A few minor changes, mainly grammatical or stylistic, are still needed:

p. 2 (of 25), last paragraph, first line: The current perception of RNA-protein interactions is strongly biased...

p. 4, 3rd paragraph, first line A protein is considered intrinsically disordered....

p. 5, bottom paragraph: These proteins contain one or two RNA binding domains at the N-terminus and a C-terminal arginine-serine rich domain...

Is this what you mean?

p. 6, 4th paragraph: disordered proteins play critical roles.

p. 9, 4th paragraph: While the numbers in Figure 2

Also: One explanation may include a protein described earlier...

p. 10, 4th paragraph: It has been shown that the CRNDE-encoding gene contains a highly conserved intron sequence...

p. 11, 4th paragraph: the assembly of the protein complex occurs
Which protein complex?

p. 12, 3rd paragraph: PRC2 was shown to bind important lncRNAs, including...
Also: ...where the appearances of low order (low-affinity binding) masks hidden complexity.

p. 12, 4th paragraph: the higher order in eukaryotic cells was proposed..... and their binding targets allow the formulation of more

p. 13, 1st paragraph: Thus, the proportion of RBPs in protein-protein interaction (PPI) networks predict if a given protein is a candidate RBP. The Yeo group...

p. 13, 3rd paragraph: From the Thomas Cech's discovery....
Also: The following sentence is ambiguous and hard to interpret. Please revise:

As depicted on Figure 3, RNA play a decisive role in the fine-tuning of expression by controlling the activity of transcription and epigenetic factors (RNA-DNA regulation) are engaged in self-control (RNA-RNA regulation; splicing, stability, translatability), but also directly regulate RBPs (RNA-protein regulation).

p. 15, Figure 3 caption: in which DNA encodes proteins...

Review form: Reviewer 3

Recommendation

Major revision is needed (please make suggestions in comments)

Are each of the following suitable for general readers?

- a) **Title**
Yes
- b) **Summary**
Yes
- c) **Introduction**
Yes

Is the length of the paper justified?

Yes

Should the paper be seen by a specialist statistical reviewer?

No

Is it clear how to make all supporting data available?

Not Applicable

Is the supplementary material necessary; and if so is it adequate and clear?

Not Applicable

Do you have any ethical concerns with this paper?

No

Comments to the Author

In this review Balcerak and coworkers have tried to highlight the different types of RNA-protein interactions and their importance. The paper discusses about the RNA-binding proteins, that are ordered, disordered and also about the RNA motifs that interact with these proteins. Though it is great to have that overall picture, it is difficult to get any detailed information when so many topics are addressed at once. In some sections there appears to be a lot of repetition of the same concept for example in the section 2.2, the intrinsically disordered proteins have been described multiple times. In other sections there is a lack of conclusive information.

I would suggest that the authors focus on one subtopic and go into the details of the findings as well as the methods that have been employed in the field, as that would convey more useful information to the readers.

Decision letter (RSOB-19-0096.R0)

21-May-2019

Dear Dr Grzybowska

We are pleased to inform you that your manuscript RSOB-19-0096 entitled "RNA-protein interactions: disorder, moonlighting and "junk" contribute to eukaryotic complexity" has been accepted by the Editor for publication in Open Biology. The reviewer(s) have recommended publication, but also suggest some minor revisions to your manuscript. Therefore, we invite you to respond to the reviewer(s)' comments and revise your manuscript.

Please submit the revised version of your manuscript within 14 days. If you do not think you will be able to meet this date please let us know immediately and we can extend this deadline for you.

- 1) A text file of the manuscript (doc, txt, rtf or tex), including the references, tables (including captions) and figure captions. Please remove any tracked changes from the text before submission. PDF files are not an accepted format for the "Main Document".
- 2) A separate electronic file of each figure (tiff, EPS or print-quality PDF preferred). The format should be produced directly from original creation package, or original software format. Please note that PowerPoint files are not accepted.
- 3) Electronic supplementary material: this should be contained in a separate file from the main text and meet our ESM criteria (see <http://royalsocietypublishing.org/instructions-authors#question5>). All supplementary materials accompanying an accepted article will be treated as in their final form. They will be published alongside the paper on the journal website and posted on the online figshare repository. Files on figshare will be made available approximately one week before the accompanying article so that the supplementary material can be attributed a unique DOI.

Online supplementary material will also carry the title and description provided during submission, so please ensure these are accurate and informative. Note that the Royal Society will not edit or typeset supplementary material and it will be hosted as provided. Please ensure that the supplementary material includes the paper details (authors, title, journal name, article DOI). Your article DOI will be 10.1098/rsob.2016[last 4 digits of e.g. 10.1098/rsob.20160049].

- 4) A media summary: a short non-technical summary (up to 100 words) of the key findings/importance of your manuscript. Please try to write in simple English, avoid jargon, explain the importance of the topic, outline the main implications and describe why this topic is newsworthy.

Images

Data-Sharing

It is a condition of publication that data supporting your paper are made available. Data should be made available either in the electronic supplementary material or through an appropriate repository. Details of how to access data should be included in your paper. Please see <http://royalsocietypublishing.org/site/authors/policy.xhtml#question6> for more details.

Data accessibility section

Sincerely,

The Open Biology Team

<mailto:openbiology@royalsociety.org>

Reviewer(s)' Comments to Author:

Referee: 2

Comments to the Author(s)

Please see the attached file (below), since it will be easier for the authors to read.

The authors, as they indicated, made most of the corrections that I had previously suggested. A few minor changes, mainly grammatical or stylistic, are still needed:

p. 2 (of 25), last paragraph, first line: The current perception of RNA-protein interactions is strongly biased...

p. 4, 3rd paragraph, first line A protein is considered intrinsically disordered....

p. 5, bottom paragraph: These proteins contain one or two RNA binding domains at the N-terminus and a C-terminal arginine-serine rich domain...

Is this what you mean?

p. 6, 4th paragraph: disordered proteins play critical roles.

p. 9, 4th paragraph: While the numbers in Figure 2

Also: One explanation may include a protein described earlier...

p. 10, 4th paragraph: It has been shown that the CRNDE-encoding gene contains a highly conserved intron sequence...

p. 11, 4th paragraph: the assembly of the protein complex occurs

Which protein complex?

p. 12, 3rd paragraph: PRC2 was shown to bind important lncRNAs, including...

Also: ...where the appearances of low order (low-affinity binding) masks hidden complexity.

p. 12, 4th paragraph: the higher order in eukaryotic cells was proposed..... and their binding targets allow the formulation of more

p. 13, 1st paragraph: Thus, the proportion of RBPs in protein-protein interaction (PPI) networks predict if a given protein is a candidate RBP. The Yeo group...

p. 13, 3rd paragraph: From the Thomas Cech's discovery....

Also: The following sentence is ambiguous and hard to interpret. Please revise:

As depicted on Figure 3, RNA play a decisive role in the fine-tuning of expression by controlling the activity of transcription and epigenetic factors (RNA-DNA regulation) are engaged in self-control (RNA-RNA regulation; splicing, stability, translatability), but also directly regulate RBPs (RNA-protein regulation).

p. 15, Figure 3 caption: in which DNA encodes proteins...

Referee: 3

Comments to the Author(s)

In this review Balcerak and coworkers have tried to highlight the different types of RNA-protein interactions and their importance. The paper discusses about the RNA-binding proteins, that are ordered, disordered and also about the RNA motifs that interact with these proteins. Though it is great to have that overall picture, it is difficult to get any detailed information when so many topics are addressed at once. In some sections there appears to be a lot of repetition of the same concept for example in the section 2.2, the intrinsically disordered proteins have been described multiple times. In other sections there is a lack of conclusive information.

I would suggest that the authors focus on one subtopic and go into the details of the findings as well as the methods that have been employed in the field, as that would convey more useful information to the readers.

Author's Response to Decision Letter for (RSOB-19-0096.R0)

See Appendix B.

Decision letter (RSOB-19-0096.R1)

29-May-2019

Dear Dr Grzybowska

We are pleased to inform you that your manuscript entitled "RNA-protein interactions: disorder, moonlighting and "junk" contribute to eukaryotic complexity" has been accepted by the Editor for publication in Open Biology.

Article processing charge

Please note that the article processing charge is immediately payable. A separate email will be sent out shortly to confirm the charge due. The preferred payment method is by credit card; however, other payment options are available.

Sincerely,

The Open Biology Team
mailto: openbiology@royalsociety.org

Appendix A

Editor's Comments to Author(s):

Although one referee is supportive, the comments of the second suggest a need to expand the review to cover the areas in more depth. If you would wish to do this, please get back to me with a plan and we can move forward

David Glover

Associate Editor,

Reviewer(s)' Comments to Author(s):

Referee: 1

Comments to the Author(s)

The authors chose a heroic topic “biology complexity” as the center discussion for this mini-review: intrinsic disordered region, moonlighting enzyme and non-coding RNA. The ambition of the authors are respectable. However, each topic has been the center of investigation of hundreds of laboratories in the world. For each topics, there has been decades of studies with the recent emerge of new findings that has shifted the paradigm. Thus, it is impossible to conduct an in-depth review on these hot topics in a short mini-review. This article glossed on the surface of these three important and sophisticated topics but didn't have a chance to provide the insight the reader would appreciate. The references in the review is mostly outdated and the figures provided are not particular original. Instead of a pan-spectrum of many mechanisms that accounts for biological complexity (which is easily a title for a multi-volume book), maybe a better strategy is to focus on one of the topics, read through the recent key findings and write a review on that.

Referee: 2

Comments to the Author(s)

This is a very good, very thorough review that merits publication in Open Biology.

I have made comments and and a large number of corrections on the original MS word manuscript. Most of these corrections are minor - mainly grammatical or stylistic. The authors should review these and make sure that my changes are consistent with what they have intended to communicate.

Authors' response:

Referee 1:

While we generally agree that the review in it's previous form could be perceived as superficial, we would like to point out that the main goal was not to comprehensively describe all of the aspects of biological complexity, but to focus on RNA-protein interactions and to describe how they contribute to complexity and on what levels. In the revised version we tried to articulate this merit more clearly. We addressed two key issues that pertain to this topic: the flexibility of RNA-protein interactions caused by structural disorder in RBPs and the variability provided by RBPs binding to non-coding RNAs. Metabolic enzymes moonlighting is treated just as a special, binary case of structural switch, providing less flexibility than

IDPs, but important for the crosstalk between cellular metabolism and the regulation of RNA. As such, we think that it cannot be omitted.

In the first version of the review these issues were joined mechanistically, without logical flow of narration and the topic of non-coding RNA was described without strict adherence to the main subject, with excessive digressions, which indeed could give an impression of glossing on the surface of every potential mechanism, without an insight.

In the revised version we provide more organized narration, pondering on previously unreported observation that while the amount of non-coding RNA is dramatically higher in more complex organisms, the number of RBPs in eukaryotes is relatively stable. This implies that the flexibility of RBPs, brought about by intrinsic disorder is crucial in managing high numbers of RNA species in eukaryotes. This observation links together the two main sources of complexity described in our review: RBPs multi-functionality and diverse functions of non-coding RNAs.

Moreover, we added a paragraph addressing the question how the plasticity and variability of RNA-protein interactions may translate into complex regulatory networks.

Additionally, we updated some of the references and modified figures, adding an original plot (Figure 2).

We feel that the revised version has been improved substantially, gained additional depth and that the corrections address this referee's comments.

Referee 2:

We included all the referee's comments, except in places where the text has been changed.

Appendix B

Authors' response:

Referee: 2

All minor changes suggested by this referee were made.

Referee: 3

Section 2.2 was carefully edited to eliminate unnecessary repetitions, however we would like to point out that what can be perceived as repetitions of the concept at the first glance, represents in fact different aspects of disordered RBPs features (physicochemical properties, induced folding phenomenon, evolutionarily conservation). To facilitate the perception of this part we have added subtitles to this section.

A sentence listing various methods used in high-throughput analyses of RNA-RBP interactions was added along with the appropriate references. However, since these methods were comprehensively described elsewhere and this review focuses on molecular mechanisms, not methods, more detailed description was omitted, since it would be introducing yet another topic, not in line with the whole narrative.

On that note, it is our conviction that reducing our review to only one subtopic (for example disorder in RBPs) will make it trivial and without the dimension build around the observation that the flexibility of RBPs, brought about by intrinsic disorder is crucial in managing high numbers of RNA species in eukaryotes.